# Hyaluronan-Loaded Liposomal Dexamethasone–Diclofenac Nanoparticles for Local Osteoarthritis Treatment

**DOI:** 10.3390/ijms22020665

**Published:** 2021-01-11

**Authors:** Ming-Cheng Chang, Ping-Fang Chiang, Yu-Jen Kuo, Cheng-Liang Peng, Kuan-Yin Chen, Ying-Cheng Chiang

**Affiliations:** 1Isotope Application Division, Institute of Nuclear Energy Research, P.O. Box 3-27, Longtan, Taoyuan 325, Taiwan; mcchang@iner.gov.tw (M.-C.C.); ckdopamine@iner.gov.tw (P.-F.C.); yjkuo@iner.gov.tw (Y.-J.K.); clpeng@iner.gov.tw (C.-L.P.); chenky@iner.gov.tw (K.-Y.C.); 2Department of Obstetrics and Gynecology, Medicine College of Medicine, National Taiwan University, Taipei 100, Taiwan

**Keywords:** liposomal nanoparticle, dexamethasone, diclofenac, osteoarthritis

## Abstract

Osteoarthritis (OA) remains one of the common degenerative joint diseases and a major cause of pain and disability in older adult individuals. Oral administration of non-steroidal anti-inflammatory drugs (NSAIDs) (such as diclofenac, DIC) or intra-articular injected gluco-corticosteroids (such as dexamethasone, DEX) were the conventional treatment strategies for OA to reduce joint pain. Current limitations for both drugs including severe adverse effects with risks of toxicity were noted. The aim of the present study was to generate a novel OA treatment formulation hyaluronic acid (HA)-Liposomal (Lipo)-DIC/DEX to combat joint pain. The formulation was prepared by constructing DIC with DEX-loaded nanostructured lipid carriers Lipo-DIC/DEX mixed with hyaluronic acid (HA) for prolonged OA application. The prepared Lipo-DIC/DEX nanoparticles revealed the size as 103.6 ± 0.3 nm on average, zeta potential as −22.3 ± 4.6 mV, the entrapment efficiency of 90.5 ± 5.6%, and the DIC and DEX content was 22.5 ± 4.1 and 2.5 ± 0.6%, respectively. Evidence indicated that HA-Lipo-DIC/DEX could reach the effective working concentration in 4 h and sustained the drug-releasing time for at least 168 h. No significant toxicities but increased cell numbers were observed when HA-Lipo-DIC/DEX co-cultured with articular chondrocytes cells. Using live-animal In vivo imaging system (IVIS), intra-articular injection of each HA-Lipo-DIC/DEX sufficed to reduce knee joint inflammation in OA mice over a time span of four weeks. Single-dose injection could reduce the inflammation volume down to 77.5 ± 5.1% from initial over that time span. Our results provided the novel drug-releasing formulation with safety and efficiency which could be a promising system for osteoarthritis pain control.

## 1. Introduction

Osteoarthritis (OA), also known as a syndrome mainly characterized by joint pain and loss of function, occurs mostly in over 45-year olds with an estimated 400 million patients suffering from arthritis worldwide [1]. At present, the mechanism of OA-induced joint degeneration is not clear. Current medical strategies cannot completely cure the pathogenic processes but only slow down the symptoms or delay the progress of the disease. OA therapeutic methods can be divided into surgical and non-surgical treatments [2]. In terms of non-surgical treatment, including rehabilitation (heat compress, electrotherapy, quadriceps strengthening), oral medications (glucosamine, analgesics), and intra-articular injections (analgesics: hyaluronic acid, platelet-rich plasma) could be considered in clinical application [3]. To eliminate joint pain and control inflammation is the major route to treat such OA inflammatory diseases.

A major strategy to reduce OA-induced inflammation and pain is by reducing the levels of prostaglandins [4]. Glucocorticoids reveal an excellent cyclooxygenase inhibition activity which mediates prostaglandins synthesis [5]. Glucocorticoids are secreted by the adrenal cortex which regulates various biochemical responses in mammals [6]. It also plays an important role in promoting cell growth and proliferation, development, and regeneration [7]. Those corticosteroid derivatives are also used as immunosuppressive agents for organ transplantation, autoimmune diseases, and anti-inflammatory reactions [8]. Among them, dexamethasone is the drug with fewer adverse side effects and wider clinical application. Thus, dexamethasone has been considered the front line drug to inhibit OA-induced inflammation [9].

Although dexamethasone has been widely used to suppress inflammation, long-term use of dexamethasone could significantly enhance drug side effects, including weight gain, lower extremity edema, bleeding tendency, poor wound healing, avascular necrosis of the humeral or femoral head, osteoporosis and fractures, muscle weakness or atrophy, gastrointestinal irritation (nausea, vomiting, ulcer) symptoms, and even pancreatitis [10,11]. To overcome the disadvantage of dexamethasone, low frequency or concentration of topical administration has been considered. However, these strategies may decrease patient compliance due to enhance severe local or systemic side effects.

The conventional corticosteroid and non-steroidal anti-inflammatory drugs (NSAIDs) formulation applied in degenerative joint illness, especially in OA diseases, have been used clinically for decades [12]. Instead of corticosteroids for OA treatment, many patients required NSAIDs for OA-induced pain relief [13]. Recent studies have suggested NSAIDs were more effective in terms of relieving pain and improving function [13]. Diclofenac is an NSAID of the phenylacetic acid class with anti-inflammatory, analgesic, and antipyretic properties [14]. Diclofenac can inhibit cyclooxygenase and reduce prostaglandin biosynthesis. It has been used for acute or chronic treatment of rheumatoid arthritis, osteoarthritis, and ankylosing spondylitis [15]. However, like other general NSAIDs, diclofenac (DIC) sodium revealed similar gastrointestinal disorders and obvious hepatocellular toxicities [16]. Thus, to develop novel dosage forms to avoid toxicities and prolong retention time in the local area will be very promising to improve the clinical application against OA inflammation and pain control.

Liposomal nanoparticles have various advantages such as increasing cellular uptake of the drugs, stabilizing physicochemicals, and the bioavailability of respective water-soluble or poorly water-soluble drugs which were significantly explored over the last decade for the drug delivery applications [17]. Electrostatic repulsion, steric repulsion, and strong hydration could affect liposomal stability [18,19]. Particle size and zeta potential are the two most important properties that determine the fate of intravenously injected liposomes. Zeta potential is the overall charge a lipid vesicle acquires in a particular medium. It is a measure of the magnitude of repulsion or attraction between particles in general and lipid vesicles in particular. Evaluation of the zeta potential of liposomal nanoparticle preparation can help to predict the stability and in vivo fate of liposomes [20]. Knowledge of the zeta potential is also useful in controlling the aggregation, fusion, and precipitation of liposomal nanoparticles, which are important factors affecting the stability of liposomal nanoparticle formulations. Several disease treatments have used the sustained-release dosage form, including cancer therapy and arthritis treatment [21,22,23,24]. In degenerative joint illness, the liposomal nanoparticle-containing drugs also have been investigated in osteoarthritis therapy for decades. However, some disadvantages such as low loading capacity, drug expulsion after crystallization, and relatively high water content of the dispersions have been observed. To generate more effective liposomal nanoparticles for OA treatment could improve the therapeutic efficiency and clinical application.

HA is a gel-like substance that is naturally present throughout the human body especially in helping in the growth and development of joint cartilage and bone [25]. HA also plays an important role in reducing joint inflammation and pain caused by injury or tissue degeneration [26]. Thus, HA intra-articular injection has become an effective therapeutic strategy for OA pain relief. Additionally, HA revealed excellent gelling properties due to its high affinity to bind water; it has become one of the most attractive controlled drug release materials in various biomedical applications [27,28]. Although both HA treatment and liposomal analgesic drugs could be effective therapeutic strategies in treating OA, the combination of HA with liposomal analgesic drugs might bring questions about new biocompatible interactions and therapeutic responses, requiring further investigations.

Liposomal dexamethasone has also been applied in phase III clinical trials recently [29]. However, there were many dimensions including drug loading capacities, releasing concentration, and time that could be improved. To improve the current sustained release formulation for OA therapeutic efficacy, we developed a hydrophilic/hydrophobic dexamethasone (DEX) with diclofenac-encapsulated liposomal-nanoparticle-containing hyaluronic acid (HA), HA-Lipo-DIC/DEX, for OA disease application. By combining the liposomal nanoparticle with HA to generate the new OA treatment formulation, HA-Lipo-DIC/DEX could not only achieve the DIC and DEX effective dose concentration in a short time but also sustained DIC- and DEX-releasing for a long period. In addition, no significant toxicities were observed when HA-Lipo-DIC/DEX co-cultured with human chondrocytes. Animal study also confirmed HA-Lipo-DIC/DEX could be more effective in suppressing arthritis in mice than untargeted liposomal DEX or free DEX. Our results provided a safe and effective new drug’s releasing formulation and it could be a promising system for OA disease treatment.

## 2. Results

### 2.1. Characteristics of Liposomal-DIC/DEX Complex (Lipo-DIC/DEX)

We synthesized various dosed ratios including DIC/DEX = 40:20 and 40:10, especially focused on different doses of hydrophilic and hydrophobic DEX as the Lipo-DIC/DEX. The Lipo-DIC/DEX nanoparticle was observed by transmission electron microscopy (TEM) (Figure 1A). TEM images from Lipo-DIC/DEX indicated that these nanoparticles had consistent sizes and were spherical as shown in dynamic light scattering (DLS) analysis (Figure 1B and Table 1). The spheroid structure revealed in the dark staining indicated the hydrophobic dexamethasone was encapsulated in the lipid bilayer of the liposome. The sizes of the various Lipo-DIC/DEX species (from 92.4 ± 0.3 to 103.6 ± 0.3 nm, respectively) did not have significant differences between each other (Figure 1C and Table 1). The majority of particles were around about 100 nm in size, demonstrating that the solutions were homogeneous with a narrow size distribution. In further experiments, encapsulation efficiency (EE) of the designed various DIC/DEX-loaded nanoparticles were examined individually. As shown in Table 1, EE of various Lipo-DIC/DEX species were all more than 90% (from 90.5 ± 5.6 to 94.7 ± 2.4%, respectively). DIC content from those liposomal nanoparticles (from 15.8 ± 2.1 to 22.5 ± 4.1%, respectively) was also similar in all formulations (Table 1). In comparison with DIC, the DEX content was significantly different based on its properties. Higher hydrophobic DEX decreased the total DEX loading percentage. DEX content in 20 mg hydrophilic DEX-loading Lipo-DIC/DEX was 4.8 ± 1.5%; however, 20 mg hydrophobic DEX-loading Lipo-DIC/DEX was only 0.9 ± 0.3%. The zeta potential of various DIC/DEX-loaded liposomal nanoparticles was from −8.8 ± 3.9 to −27.9 ± 5.4 mV, respectively (Table 1).

These results indicated that all designed nanoparticles were successfully prepared in the nanoscale; In compared with the 40 mg DIC combined with 20 mg DEX-loaded group, the 40 mg DIC combined with 5 mg hydrophilic/5 mg hydrophobic DEX group were superior in encapsulation efficiency and drug content. We then used 40 mg DIC combined with 5 mg hydrophilic/5 mg hydrophobic DEX-loaded liposomal nanoparticles as the standard recipe for further studies.

### 2.2. Lipo-DIC/DEX Releasing Profiles

We next investigated in vitro DIC and DEX releasing from Lipo-DIC/DEX. The release profile of the Lipo-DIC/DEX nanoparticle was carried out by dialysis bag diffusion method at 37 °C in phosphate-buffered saline (PBS). It took less than 4 h to reach an effective dose of DIC (0.00015–0.105 mg/mL, recommended doses) in the Lipo-DIC/DEX formulation. The time–DIC releasing correlation was as described: control, −0.000 ± 0.000 mg/mL; 4 h, 0.045 ± 0.003 mg/mL; 6 h, 0.069 ± 0.005 mg/mL; 18 h, 0.071 ± 0.003 mg/mL; 24 h, 0.078 ± 0.003 mg/mL; 48 h, 0.076 ± 0.003 mg/mL; 72 h, 0.080 ± 0.005 mg/mL; 96 h, 0.080 ± 0.05 mg/mL; 120 h, 0.080 ± 0.005 mg/mL; 144 h, 0.083 ± 0.006 mg/mL; and 168 h, 0.084 ± 0.007 mg/mL (Figure 2A).

The time–hydrophilic DEX releasing profile from Lipo-DIC/DEX was also measured as described: control, −0.000 ± 0.000 mg/mL; 4 h, 0.010 ± 0.000 mg/mL; 6 h, 0.015 ± 0.000 mg/mL; 18 h, 0.026 ± 0.003 mg/mL; 24 h, 0.027 ± 0.003 mg/mL; 48 h, 0.028 ± 0.003 mg/mL; 72 h, 0.026 ± 0.004 mg/mL; 96 h, 0.025 ± 0.004 mg/mL; 120 h, 0.024 ± 0.003 mg/mL; 144 h, 0.025 ± 0.003 mg/mL; and 168 h, 0.024 ± 0.004 mg/mL (Figure 2B). The hydrophobic DEX-releasing concentration was approximately one-fifth compared to the hydrophilic DEX. The releasing profiles were 4 h: 0.003 ± 0.000 mg/mL; 6 h: 0.004 ± 0.000 mg/mL; 18 h: 0.005 ± 0.001 mg/mL; 24 h: 0.004 ± 0.000 mg/mL; 48 h: 0.005 ± 0.000 mg/mL; 72 h: 0.003 ± 0.001 mg/mL; 96 h: 0.003 ± 0.001 mg/mL; 120 h: 0.003 ± 0.001 mg/mL; 144 h: 0.003 ± 0.001 mg/mL; and 168 h: 0.003 ± 0.001 mg/mL (Figure 2B).

These results revealed that DIC and hydrophilic and hydrophobic DEX encapsulated liposomal nanoparticles could generate an increasing drug concentration. The Lipo-DIC/DEX could not only achieve the DIC and DEX effective dose concentration in a short time but also sustained DIC and DEX release for a long period.

### 2.3. HA-Lipo-DIC/DEX Release Profiles

HA is a relatively new OA treatment that has shown varied results through several clinical trials [30,31]. It can be used as a scaffold for engineering new treatments and several new preparations for OA treatment formulations. To develop a new effective DIC/DEX formulation for ophthalmic application, we combined the HA complex with Lipo-DIC/DEX to form HA-Lipo-DIC/DEX, which naturally occurs within the cartilage and synovial fluid loaded with liposomal DIC and hydrophilic/hydrophobic DEX, which was formulated to sustain releasing for at least one week. We then investigated the release of both DIC and DEX from HA-Lipo-DIC/DEX. Figure 2 illustrates the results of the investigation of DIC- and DEX-release from HA-Lipo-DIC/DEX. As shown in Figure 2, the release of DIC (Figure 2C) and DEX (Figure 2D) from HA-Lipo-DIC/DEX resulted in an effective dose in 4 h (the timepoint–drug release correlation ranged from 0.039 ± 0.003 to 0.088 ± 0.003 mg/mL for DIC, Figure 2C), 6 h (the timepoint–drug release correlation ranged from 0.015 ± 0.002 to 0.033 ± 0.003 mg/mL at each timepoint for hydrophilic DEX, Figure 2D), and 18 h (the timepoint–drug release correlation ranged from 0.003 ± 0.000 to 0.005 ± 0.001 mg/mL at each timepoint for hydrophobic DEX, Figure 2D), respectively.

Based on the results presented in Figure 2C,D, we confirmed that the release of both DIC and DEX from HA-Lipo-DIC/DEX could be sustained stably. This formulation could not only generate an effective dose/concentration of DIC and DEX in a short time but could also sustain the release of DIC and DEX for a long period.

### 2.4. Biocompatibility Analysis of HA-Lipo-DIC/DEX

Our previous experiments have revealed the properties of HA-Lipo-DIC/DEX in terms of rapidly generating an effective dose/concentration and sustaining the stable release of DIC/DEX for a long period. We also investigated the biocompatibility of HA-Lipo-DIC/DEX and whether this novel OA therapeutic formulation produced chondrocytes toxicity. Consequently, in the current study, we co-cultured chondrocytes with either free liposome, free DIC combined with hydrophilic/hydrophobic DEX or HA-Lipo-DIC/DEX, and then calibrated the viable cells to analyze the potential toxicity. Cells counts from the co-culture with HA for 24, 48, and 72 h were defined as the control (100%, Figure 3). As shown in Figure 3, the cell survival rate especially in the liposome only (92.74 ± 1.79%) and HA-Lipo-DIC/DEX-treated groups (93.65 ± 3.77%) were significantly higher compared to free DIC combined with the hydrophilic/hydrophobic DEX-treated group (65.48 ± 4.01%, 24 h after 108.8 mg DIC/37.5 mg DEX treatment, Figure 3A). There was no significant difference between liposome only and HA-Lipo-DIC/DEX-treated groups. In addition, the cell survival rate after drug treatment for 48 h and 72 h of the liposome only (48 h: 90.44 ± 2.59%; 72 h: 81.96 ± 4.02%) and HA-Lipo-DIC/DEX-treated groups (48 h: 91.68 ± 5.34; 72 h: 86.03 ± 4.06%) remained superior to the free DIC combined with hydrophilic/hydrophobic DEX-treated group (8 h: 50.98 ± 2.24; 72 h: 57.44 ± 0.15%, Figure 3B,C).

This evidence suggests that HA-Lipo-DIC/DEX did not induce significant toxicity in chondrocytes even in high dose treatments. Moreover, the sustained releasing DIC and DEX without cellular toxicity might help in inhibiting OA-inducing inflammation and pain control.

### 2.5. Effects of HA-Lipo-DIC/DEX in Suppressing Arthritis in Mice

The previous evidence that we have presented of HA-Lipo-DIC/DEX treatment could not induce chondrocytes toxicity, and the long-term drug releasing property could sustain more than seven days. To validate these findings in vivo, we topically administered exogenous free DIC, free DEX, or HA-Lipo-DIC/DEX into sub plantar tissue of the right hind paw of Freund’s Complete Adjuvant (FCA)-treated mice in order to observe the anti-inflammation and relative efficacy of the liposomal nanoparticle formulation. As shown in Figure 4, HA-Lipo-DIC/DEX was revealed more efficacious in inhibiting hind paw thickness than either free DIC or free DEX 14 days after treatment. The paw thickness of each group at day 28 after FCA treatment was as indicated: FCA only, 5.29 ± 0.26 mm; free DIC, 4.60 ± 0.26 mm; free DEX, 3.73 ± 0.24 mm; and HA-Lipo-DIC/DEX, 2.74 ± 0.14 mm, respectively. These results indicate that both free DIC and DEX treatment could inhibit inflammation in vivo; however, HA-Lipo-DIC/DEX could significantly inhibit OA-induced inflammation in comparison with other groups in vivo.

### 2.6. Decreased Fluorescence Signals of Neutrophil Elastase in OA-Induced Mice after HA-Lipo-DIC/DEX Treatment

Previous investigations have demonstrated HA-Lipo-DIC/DEX could inhibit FCA-induced hind paw inflammation. To validate these findings in a pathogenic animal model, we topically administered exogenous HA-Lipo-DIC/DEX into the right hind paw then evaluated the fluorescence emission in murine arthritic joints after local administration of HA-Lipo-DIC/DEX by using in vivo imaging. Figure 5A shows the fluorescence intensities in arthritic paws after administration of FCA for 7 days without any treatment and followed by HA-Lipo-DIC/DEX treatment for another 14 days, respectively. As shown in Figure 5, the fluorescence intensities of right hind paws were lowest in mice in the HA-Lipo-DIC/DEX group compared to those in the other groups (Figure 5B) (*p* < 0.001, by one-way ANOVA). These results indicate that HA-Lipo-DIC/DEX could significantly inhibit neutrophil elastase and inflammation in vivo.

### 2.7. HA-Lipo-DIC/DEX Treatment Reduced Arthritic Inflammation and Leukocytes Infiltration

We subsequently determined if there were any differences in the levels of infiltration of leukocytes or lymphocytes within the different arthritic paws samples treated with the various HA-loaded DIC/DEX liposomal nanoparticles. Tissue blocks were prepared from mice administrated with free DIC, free DEX, or HA-Lipo-DIC/DEX as described earlier. The representative figures of the infiltrating leukocytes within the arthritic paws of the different treatment groups evaluated by hematoxylin and eosin staining are presented in Figure 6A. The arthritic paws without any treatment exhibited increased inflammation and also edema progression. In comparison with the untreated group, all anti-inflammation drug treatment groups, especially the HA-Lipo-DIC/DEX-treated group, exhibited not only significantly suppress edema.

Histological scoring including severity of inflammation, cartilage destruction, and bone erosion of hind paw from OA-induced mice after various recipe treatments were also scored (Figure 6B). The score of infiltrating leukocytes was reduced in the paws treated with free DEX (2.7 ± 0.6), especially in HA-Lipo-DIC/DEX (2.3 ± 0.3), in comparison with the control (3.0 ± 0.0) or free DIC (3.0 ± 0.0) administration locally. However, mice receiving free DEX administration revealed significant bone erosion (1.7 ± 0.6) and loss of superficial chondrocytes (1.3 ± 0.6) in the articular cartilage. On the other hand, the HA-Lipo-DIC/DEX-treated group showed moderate mononuclear cell infiltration in the periosteal/periarticular soft tissues and slight bone erosion (1.3 ± 0.6). This evidence indicates that the HA-Lipo-DIC/DEX formulation produces a long-term drug-releasing property that enhanced a strong anti-inflammatory effect, thereby maintaining OA-induced pain relief.

## 3. Discussion

DIC is a non-steroidal compound with significant anti-rheumatic, anti-inflammatory, analgesic, and antipyretic effects that can be used to treat rheumatic or non-rheumatic inflammatory pain, and even post-operation pain control [32]. Oral administration of DIC could be rapidly absorbed but only about 50% could be metabolized and available systemically [33]. Following treatment, DIC binds extensively to plasma albumin and is attained in synovial fluid with substantial concentrations [33]. There are two major routes to metabolite diclofenac in humans: the acyl glucuronidation which could be catalyzed by uridine 5’-diphosphoglucuronosyl transferase 2B7 and phenyl hydroxylation catalyzed by cytochrome P4502C9 and 3A4 [34]. Whether route the DIC is metabolized through, the end product is highly related to its formation concentration. The clearance of DIC product is depending on its biotransformation to glucuro-conjugated and sulfate metabolites [34]. Further hydroxylation of diclofenac glucuronide was shown to occur in vitro with recombinant CYP2C8, which may be of clinical significance in terms of defining major metabolic routes involved in the elimination of diclofenac in humans [35]. Approximately 65% of an oral dose of DIC is filtered through the glomerulus and discharged into the urine and 35% as bile conjugates [34]. DIC has generally been shown to be relatively safe and well tolerated compared to other nonselective NSAIDs (e.g., naproxen and ibuprofen); however, some significant safety risks such as gastropathy potential remain a consideration.

DIC is generally well tolerated. Its most common side effects include abdominal pain, nausea, vomiting, headache, and constipation; however, these symptoms are usually mild [36]. Thus, DIC is an important treatment option for pain remission. A recent study for OA patients treated with NSAIDs showed approximately 86.6% increased gastrointestinal risk and 44.2% of patients were at high cardiovascular risk [37]. Other investigations also illustrated NSAIDs are associated with increased risk of hepatic and renal toxicity [38]. With the advancement of medical research, it was made possible to develop DIC topical formulations instead of oral administration to reduce incidence of these adverse effects. Topical diclofenac formulation offers patients with osteoarthritis of the knee a safer therapeutic option than oral diclofenac or other NSAIDs. However, Chang and his colleagues also indicated the therapeutic concentrations of non-selective NSAIDs including diclofenac could inhibit thymidine incorporation and cell cycle arrest that might cause proliferation suppression and cell death of chondrocytes [39]. These issues led to limitation in those NSAIDs’ intra-articular injection treatment. Developing a sustained releasing formulation of diclofenac could be a useful resolution for reducing those adverse effects. Elron-Gross et al. have developed therapeutic liposome diclofenac formulations for OA treatment previously [40]. Our study was also focused on the development of new OA-treatment formulations to improve patient quality of life. Based on the results of the current study, we developed a novel DIC/DEX-containing OA-therapeutic formulation, HA-Lipo-DIC/DEX (Figure 7), which exhibited that it could release an effective concentration of the drugs in a short time and facilitate the sustained release for a long period (Figure 2). A biocompatibility analysis revealed that this new drug formulation did not result in significant cytotoxicity (Figure 3). Thus, the subsequent development of HA-Lipo-DIC/DEX has the potential to become the new therapeutic strategy for OA-induced disorders including for inflammation and pain.

The adverse effects of oral corticosteroids are widely recognized including stomach upset, headache, dizziness, menstrual changes, trouble sleeping, increased appetite, or weight gain [41]. Questions remain about how well their safety has been issued in populations who had to use this effective anti-inflammation drug in the long term. To avoid those unwanted effects enhanced by oral corticosteroids, current recommendations for patients with knee OA, particularly with signs of local inflammation with joint effusion, are to use topical corticosteroid injections. Steroid injections into joints can be a useful way to help to manage joint pain for people with conditions such as osteoarthritis for years [42]. However, Wernecke et al. also demonstrated high doses of corticosteroids were associated with significant gross cartilage damage and chondrocyte toxicity [43]. Thus, corticosteroid-loaded liposomes are currently being evaluated in clinical applications. Here, we comprehensively developed the hydrophilic/hydrophobic DEX containing liposomal nanoparticle HA-Lipo-DIC/DEX to investigate the potential application in OA joint pain treatment. Evidence indicated the liposomal DEX-containing formulation significantly decreased chondrocyte toxicity both in vitro (Figure 3) and in vivo (Figure 5). Furthermore, leukocyte infiltration was also reduced after HA-Lipo-DIC/DEX intraarticular injection (data not shown). The findings indicate that the HA-Lipo-DIC/DEX modulates their cellular mechanism of action and provides important indications for further investigations.

Liposomal nanoparticles have improved the therapeutic and delivery efficiency of various formulations and have been used in clinical applications for decades [44,45]. In this study, a formulation consisting of HA combined with liposomal DIC/DEX nanoparticles was developed and tested as an OA potential anti-inflammation and pain relief treatment. The ideal liposomal nanoparticle with optimum stability retained drug delivery efficacy and encapsulation without loss before drug arrival at the target organ. There were various neutral phospholipids including di-palmitoylphosphatidylcholine (DOPC), 1,2-distearoyl-sn-glycero-3- phosphocholine (DSPC), dimyristoyl-phosphatidylcholine (DMPC) and dipalmitoyl phosphatidylcholine (DPPC) could be designed and prepared as liposomal nanoparticles for drug delivery systems. DOPC is a zero-curvatured lipid and forms a stable liposomal bilayer when used alone. When DOPC has been formulated as the emulsion, the clearances were slower but the drug loading capacity was more effective in comparison to other high-density lipoprotein fractions. In this study, we also prepared DOPC for use as an emulsifying agent in order to improve the encapsulation and emulsification efficiency, as described previously. Fernandes and his colleagues have also demonstrated that the encapsulation of DIC was more effective in DOPC liposomes compared to in other phosphatidylcholine liposomes and that DIC could be displaced from the liposomes as the cholesterol content of the liposome membranes was increased [46]. In addition, owing to the different drug delivery route, the new drug formulation developed in the current study will not directly enter the blood vessels and cause vascular occlusion.

OA models have been largely developed to reflect pathologic properties as the human OA pain models. Previous investigations have developed several models including Dunkin Hartley guinea pigs or STR/ort mice [47]. However, those models could not perform an effective or accurate disease progression. Since OA is a chronic disease with acute inflammatory pain models, lacking those pathogenic properties might limit translational relevance to human OA. The ideal established OA models should display some characteristics of humans’ symptoms such as osteophytes, cartilage damage, subchondral bone remodeling, and pain behavior. Adjuvant- and collagen-induced arthritis models revealed a short duration of testing, easy measurement, and similarities to human OA, thus, have been used to study the mechanisms underlying joint pain and may have relevance to aspects of OA pain [48]. Our results of the current animal model were also performed by mono-articular injection of FCA (Figure 4 and Figure 5). Poly-articular arthritis could be further generated by systemic FCA or collagen injection. Base on the proof-of-concept, our new analgesics formulation HA-Lipo-DIC/DEX was initially used as single joint pain relief. Further, we used the degrees of paw swelling (Figure 4) to simulate the severe pain and debilitation of a human end point. Significant anti-inflammation could be also measured after we treated OA mice with HA-Lipo-DIC/DEX. Further investigations especially focused on poly-articular arthritis will be conducted in sequential studies in order to expand the applications of this novel slow-releasing pharmaceutical formulation.

OA is most commonly a result of aging cartilage. Aging cartilage progressively becomes vulnerable and stiffer. Treatment of OA is aimed at managing symptoms, restoring/maintaining function, and slowing the progression of arthritis. Our investigations were focused on developing a biodegradable material containing HA to be used as a scaffold for the liposomal nanoparticles to maintain sustained releasing properties in an arthritis cavity [49]. Previous studies have also demonstrated that similar materials and formulations with high biocompatibility may be a promising solution for use in tissue engineering applications [49,50]. However, by combining different polarities of hydrophilic/hydrophobic DEX with DIC, our HA-Lipo-DIC/DEX could release effective DEX and DIC for a longer time (Figure 3). In addition, the composition of HA from HA-Lipo-DIC/DEX could refill the joint lubrication and viscoelastic properties of the synovial fluid in the affected joint. The results of both in vitro and in vivo experiments have confirmed that HA-Lipo-DIC/DEX could not only generate an effective dose/concentration of DIC/DEX in a short time but could also sustain the DIC/DEX release for a long period (Figure 3 and Figure 5). Additionally, owing to its high biocompatibility, the liposomal nanoparticle formulation could inhibit the inflammation status and paw swelling. Our formulation has the potential not only to enhance the medication adherence and quality of life of patients but also to increase therapeutic efficiency.

Overall, the results of our current investigation on the efficacy of the new liposomal formulation to induce anti-inflammation effects demonstrate its potential application for OA-induced pain control. The beneficial effects of strategies based on similar mechanisms have been predominantly reported in the case of rheumatoid-arthritis-targeted therapy. A biodegradable matrix involving liposomal analgesics’ administration to the articular cavity has been found to be an appropriate option for treating cases, especially for OA. Our novel ophthalmic formulation could minimize the adverse responses that lead to acute analgesics cytotoxicity in tissue that is not harmed. However, there still are several challenges associated with the translation of this treatment modality to human clinical applications. A key issue is that the biomatrix should degrade at its setting time to reduce discomfort in order to enhance patient compliance. Moreover, the optimal dose, fraction, frequency, and timing of treatment must be optimized based on disease severity and location. Further research is warranted to facilitate the future application of this type of treatment.

## 4. Conclusions

In the development of OA analgesics formulations, the use of liposomal nanoparticles containing hydrophilic/hydrophobic DEX with DIC that control inflammation by the sustained releasing of an effective analgesics concentration has attracted considerable attention. Via the preparation of a biocompatible HA hydrogel, a decomposition formulation was developed comprising a novel formulation with liposomal nanoparticles containing analgesics that released an effective working drug concentration in a short time, and the drug release could be sustained for 168 h. The newly developed OA analgesics formulation did not exhibit any cytotoxic properties. An animal study also confirmed that this novel OA analgesics formulation could effectively inhibit inflammation and paw swelling. Thus, the novel drug-releasing formulation provided safety and efficiency that has great potential as the future treatment strategy of OA pain control.

## 5. Materials and Methods

### 5.1. Cell Culture

Human chondrocytes SW 1353 and culture reagents were all purchased from the ATCC (Manassas, VA, USA). Cells were maintained in fresh Leibovitz’s L-15 Medium (ATCC) supplemented with 10% fetal bovine serum, L-glutamine (6 mM), and 10 U/mL penicillin/10 µg/mL streptomycin in a CO_2_ incubator at 37 °C and with 5% CO_2_. The cells were sub-cultured by trypsin (0.25%; Invitrogen, Carlsbad, CA, USA) and dissociated at approximately 80% confluence. The culture media were changed every 2–3 days.

### 5.2. Material Preparation

Hyaluronic acid (molecular weight from 130,000–150,000), hydrophilic and hydrophobic dexamethasone, diclofenac, Di-palmitoylphosphatidylcholine (DOPC), cholesterol, DIC and DEX were all purchased from Sigma-Aldrich (St. Louis, MO, USA).

### 5.3. Preparation of DIC- and DEX-Containing Liposome (Lipo-DIC/DEX)

The protocol for preparing liposomal antibiotics was as described previously [51]. Briefly, 5 mg hydrophobic DEX combined with 7.7 mg cholesterol and 110.6 mg DOPC (molar ratio = 14:15:2) were solved in 5 mL methylene chloride. The solvent was evaporated in a rotary evaporator at 50 °C, and solvent traces were removed in a 60 °C oven for a further 30 min. The phospholipid film was hydrated with 5 mg hydrophilic DEX and 40 mg DIC-containing double-distilled water using a probe-type sonicator at 35 W for 30 min for Lipo-DIC/DEX self-assembled. The liposomal system was further filtered by a 0.2 μM filter and purified by a PD-10 column within 50 °C phosphate-buffered saline (PBS) to deplete the unencapsulated DIC and DEX in the supernatant.

### 5.4. Preparation of HA-Lipo-DIC/DEX

HA-Lipo-DIC/DEX was first to be determined for OA formulations. To generate HA-Lipo-DIC/DEX, Lipo-DIC/DEX was mixed with an equal volume of liquefied HA (20 mg/mL) solution at 50 °C follow by standing at room temperature for 2 h. The final liquefied HA concentration was determined as 10 mg/mL.

### 5.5. Microscopic Analysis and Transmission Electron Microscopy

Transmission electron microscopy (TEM) studies were performed to establish direct visual monitoring of individual particle sizes of manufacturing liposomes. Briefly, a drop of a sample was placed onto a carbon-coated copper grid (Polysciences Inc., Warrington, PA, USA) along with a drop of 2% phosphotungstic acid to leave a thin film and the excess solution was then drained off using filter paper. Negative staining of samples was also employed by using 2% *w/v* aqueous uranyl acetate solution. The grid was allowed to air dry thoroughly, and the samples were then viewed using a Hitachi transmission electron microscope (Hitachi H-7650, Tokyo, Japan). TEM images and electron diffraction patterns were visualized and collected by soft imaging software.

### 5.6. Zeta Potential and Particles Size Analysis of Liposomes

The complete liposomes with negative surface charges could inhibit the aggregation or surface repulsion due to similarly charged particles [51,52]. Zeta potential indicates the overall charge present and stability of liposomal systems. The mean size and polydispersity index (P.I.) were calculated by Malvern Zetasizer Nano ZS90 (Malvern Instruments Ltd., Malvern, UK). The zeta potential values were calculated from the electrophoretic mobility by means of the Helmholtz–Smoluchowski relationship [53]. Zeta potential measurement is also important to monitor liposomal stability during liposomes’ precipitation and fusion control [54].

The size and surface charge of DIC- and DEX-loaded liposomes were measured by dynamic light scattering (DLS; Delsa™ Nano Particle Analyzer; Beckman Coulter, Fullerton, CA, USA) and zeta potential (Coulter DELSA 440 SX), respectively. The autocorrelation function of the scattered light was analyzed by the cumulant method as described previously [55].

### 5.7. The Measurement of Loading Capacity (LC) and Encapsulation Efficiency (EE)

Both LC and EE indicated the quantity of drugs entrapped in liposomal formulations. EE was especially important; it can be used to optimize the formulation composition during studying these entrapped agents in either physical or biological systems. The content of entrapped DIC or hydrophilic or hydrophobic DEX was measured by high-performance liquid chromatography (HPLC, MyBioSource, San Diego, CA, USA). Separation was achieved on a Waters Symmetry 300 C18 reversed-phase column (25 cm × 4.6 mm, 5 mm) using 0.1% trifluoroacetic acid (TFA) (*w/v*) in acetonitrile (50:50, *v/v*) as the mobile phase and a flow rate of 1 mL/min. DIC or hydrophilic or hydrophobic DEX was detected by fluorescence at excitation and emission wavelengths of 254 nm and 280 nm, respectively. Quantification was achieved by comparing the mean peak areas of samples from tissues spiked with known amounts of DIC or hydrophilic or hydrophobic DEX.

The loading capacity (LC%) and encapsulation efficiency (EE%) were derived by taking the equations below (1 and 2), respectively.
LC% = (Total amount of determined DIC or DEX/Total amount of dried liposome) × 100%(1)
EE% = (Total amount of determined DIC or DEX/Initial amount of DIC or DEX loading) × 100%(2)

### 5.8. Measurement of In Vitro Release

The in vitro release profiles of DIC or DEX from the HA-Lipo-DIC/DEX were studied as described previously [51,56]. Briefly, 2 mL volume of HA-Lipo-DIC/DEX were placed in a dialysis bag (molecular weight cutoff 3.5 kDa), and the dialysis bag was incubated in 50 mL of PBS (pH ~ 7.4) with gentle stirring at 37 °C, and the release medium was collected at predetermined time intervals. To detect the released DIC and DEX, the samples were quantified by HPLC according to the manufacturer’s instruction as described above. The samples were run in triplicate and the mean values were calculated. All results are the mean ± standard deviation.

### 5.9. Induction and Evaluation of OA in C57BL/6 Mice

Five- to six-week-old male C57BL/6 mice were purchased from the National Laboratory Animal Center (Taipei, Taiwan) and bred in the animal facility of INER (Taoyuan, Taiwan). All animal procedures were performed following approved protocols that were developed in accordance with recommendations for the proper use and care of laboratory animals (approved number: 109022). The mice were kept at 21–23 °C at a light–dark cycle of 12 h. For arthritis induction, mice were anesthetized with isoflurane, and mineral oil was injected in the right ankle joint for the control group of animals. Adjuvant arthritis was induced by subcutaneous injection of 0.1 mL of Freund’s Complete Adjuvant (FCA, Sigma, St. Louis, MO, USA) into sub plantar tissue of the right hind paw of each mouse [57]. The test groups consisted of FCA-injected mice challenged with the respective doses of the test drugs administered locally. The mice were then observed regularly for signs of arthritis. The disease severity was evaluated on the basis of erythema and swelling of the paws.

### 5.10. DEX (Liposomal/Free) Treatment of Mice

Arthritis was induced in mice by injection of FCA subcutaneously as described above. FCA-treated mice were randomized and assigned to four groups (*n* = 3 per group) for local injection (100 μL/mice) of PBS, free DIC, free DEX, and HA-Lipo-DIC/DEX. These injections were performed three days after FCA injection. The dose of DEX was 0.1 mg/kg body weight. Mice were then examined regularly and graded for arthritic severity as described above.

### 5.11. Proteolytic Activity of Neutrophil Elastase in FCA-Inflamed Knee Joints

Neutrophil elastase was secreted by neutrophils and macrophages which have been used to determine the chronic inflammation status [58]. In the current study, the proteolytic activity of neutrophil elastase in the inflamed knee joints was determined by using the substrate Neutrophil Elastase 680 FAST (NE 680, PerkinElmer, Waltham, MA, USA), a commercially available fluorescence agent that is optically silent upon injection and produces a fluorescence signal after specific cleavage by neutrophil elastase on days 1 and 14 post-FCA injection. Briefly, mice were anesthetized (2–4% isoflurane; 100% oxygen at 1 L/min) and NE680 (1 nmol/25 μL) was injected intravenously. Four hours after NE680 injection, mice were placed in a prone position in the imaging chamber of an In-Vivo Xtreme imaging system (Bruker Corporation, Billerica, MA, USA). The substrate was excited at a 650 nm wavelength and the emitted fluorescence was captured at a 700 nm wavelength for a 2.5-s exposure time. For analysis, identical regions of interest were drawn around both the ipsilateral (arthritic) and contralateral (naïve) joint, and fluorescence intensities were calculated for each knee.

### 5.12. Histopathological Examination

The ankle joint specimens from the mice treated with PBS, free DIC, free DEX, or HA-Lipo-DIC/DEX were fixed in 10% buffered neutral formalin and were embedded in paraffin. Sections were cut at a thickness of 3–5μm and were stained with hematoxylin and eosin. The histopathological changes, including cell morphology and the presence of metastatic tumor cells, were examined by light microscopy (40× and 200×).

### 5.13. Histological Scoring System for Arthritis

The scoring system for arthritis was based on the severity of lesions. The grading scale of 0–3 according to the proportion of areolar tissue infiltrated with mononuclear cells was as described previously [59]. Synovial lining cell hyperplasia and the number of histiocytes observed in the synovial tissue were graded similarly on the same scale. Cartilage destruction was scored on a scale of 0–3, ranging from no damage to fully destroyed cartilage layers. Bone erosion was scored according to the following features: 0 = normal, 1 = mild loss of cortical bone at a few sites, 2 = moderate loss of cortical trabecular bone, and 3 = marked loss of bone at many sites. From every joint (3 mice per group), 4 or 5 sections were taken at different depths to provide a representative sample of the whole joint. Mean scores were determined from the different sections of the individual animals, and composite scores for the different experiments.

### 5.14. Statistical Analysis

Statistical analyses were carried out using SPSS version 6.0 for Windows, which involved a one-way ANOVA and the Mann–Whitney *U* test. A *p* value *<* 0.05 was considered to be statistically significant.

## Figures and Tables

**Figure 1 ijms-22-00665-f001:**
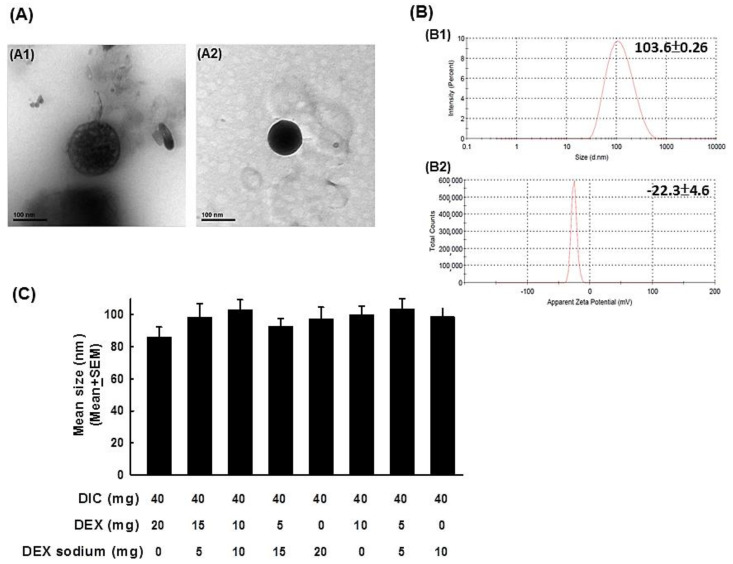
Characteristics of various dosages of DIC/DEX-loaded liposomes. (**A**) Representative figures of DIC/DEX-loaded liposomal nanoparticles captured by transmittance electron microscopy. The left panel depicts an empty liposomal nanoparticle. The right panel depicts a liposomal nanoparticle that was loaded with DIC/DEX. (**B**) The representative figures of the size distribution and zeta potential in 40 mg/mL DIC combined with 5 mg/mL hydrophilic/hydrophobic DEX-loaded liposomal nanoparticles acquired by DLS analysis (B1: size distribution, B2: zeta potential). (**C**) Bar figures of size distribution of various dosages of DIC/DEX-loaded liposomal nanoparticles. Note: The average sizes of the DIC/DEX-loaded liposomal nanoparticles were approximately 200-400 nm. Abbreviations: DLS, dynamic light scattering; DIC, diclofenac; DEX, dexamethasone.

**Figure 2 ijms-22-00665-f002:**
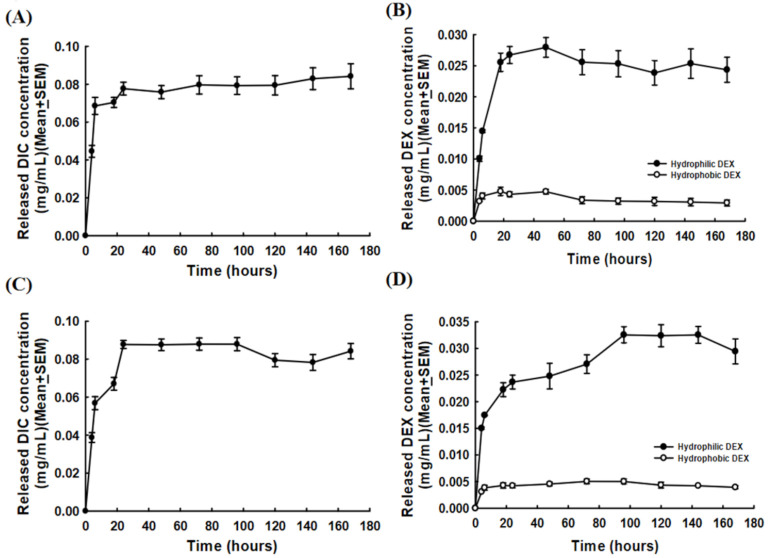
The release profiles of various Lipo-DIC/DEX formulations in vitro. (**A**) DIC release concentrations from Lipo- DIC/DEX at different timepoints. Data were expressed as mean ± SEM. (**B**) DEX release concentrations from Lipo-DIC/DEX at different timepoints. (**C**,**D**) DIC/DEX release concentrations from HA-Lipo-DIC/DEX at different time points were also measured. Data were expressed as mean ± SEM. Abbreviations: Lipo, liposomal; DIC, diclofenac; DEX, dexamethasone; HA, hyaluronic acid; SEM, standard error of the mean.

**Figure 3 ijms-22-00665-f003:**
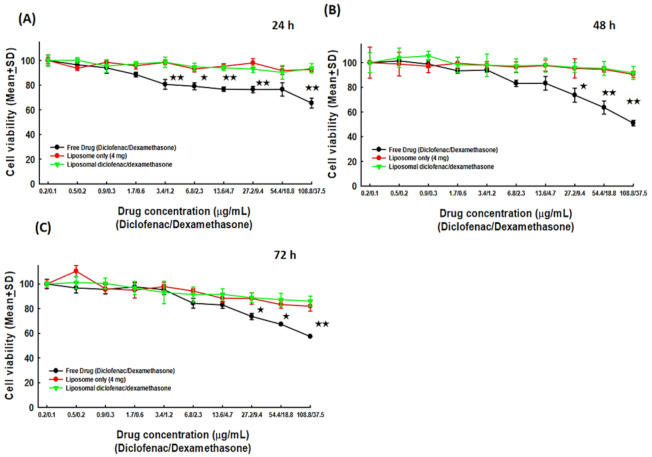
Biocompatibility analysis of HA-Lipo-DIC/DEX. Bar figure of the viability of human chondrocytes SW 1353 co-cultured with HA-Lipo-DIC/DEX. (**A**) SW 1353 chondrocytes co-cultured with various drug concentration ratio for 24 h. (**B**) SW 1353 chondrocytes co-cultured with various drug concentration ratio for 48 h. (**C**) SW 1353 chondrocytes co-cultured with various drug concentration ratio for 72 h. SW 1353 chondrocytes viability were significantly higher in co-culturing with liposome only, and HA-Lipo-DIC/DEX in compared with free drugs. (^★^ indicates *p* < 0.05, ^★★^ indicates *p* < 0.01, one-way ANOVA). Abbreviations: HA, hyaluronic acid; Lipo, liposomal; DIC, diclofenac; DEX, dexamethasone.

**Figure 4 ijms-22-00665-f004:**
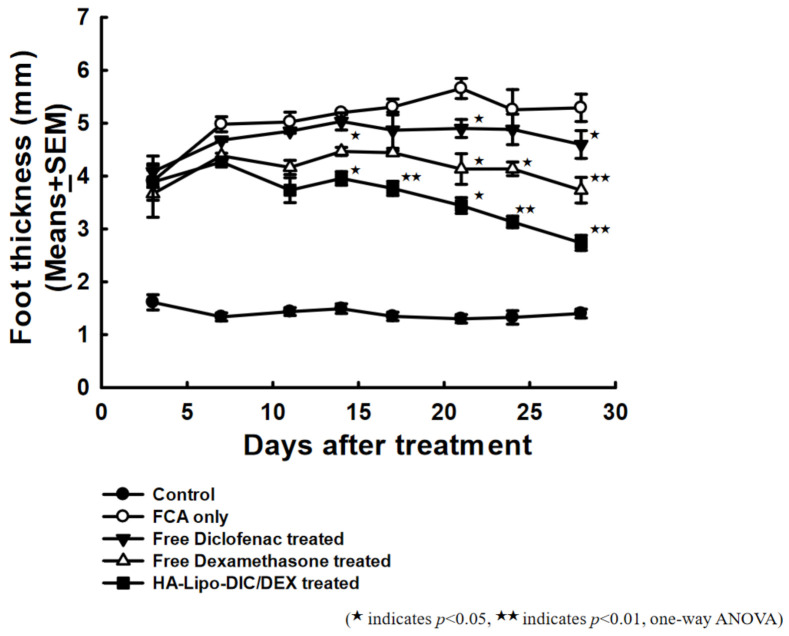
Comparative efficacy of HA-Lipo-DIC/DEX and free drugs for arthritis therapy. Arthritis was induced in C57BL/6 mice by subcutaneous injection of FCA as described in Materials and Methods. At the time of appearance of signs of arthritis (disease onset), mice were randomized (*n* = 5 per group) and given an intraarticular injection single dose from day eight of the following preparations: FCA only, free DIC, free DEX, or HA-Lipo-DIC/DEX. The opposite paw was served as the reference for comparison with other groups. (^★^ indicates *p* < 0.05, ^★★^ indicates *p* < 0.01, one-way ANOVA). Abbreviations: FCA, Freund’s Complete Adjuvant; HA, hyaluronic acid; Lipo, liposomal; DIC, diclofenac; DEX, dexamethasone.

**Figure 5 ijms-22-00665-f005:**
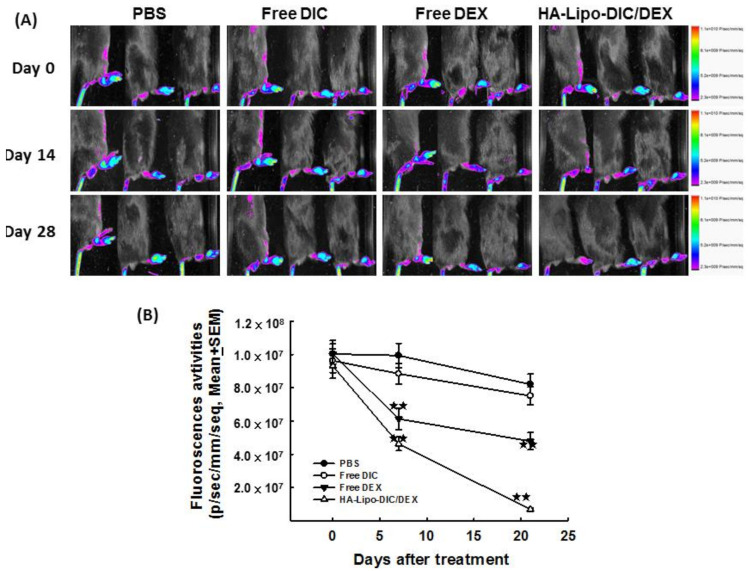
In vivo imaging of systemically administered liposomes. Arthritis was induced in C57BL/6 mice by subcutaneous injection of FCA as described in Materials and Methods. At the time of appearance of signs of arthritis (disease onset), mice were randomized (*n* = 5 per group) and given intravenously NE680 (1 nmol/25 μL) two injections every 14 days starting from day 7 (indicated by arrows) to monitor the inflammation status. Mice then receiving single-dose treatment from day 8 of the following preparations: FCA only, free DIC, free DEX, or HA-Lipo-DIC/DEX. (**A**) The fluorescence imaging of arthritic mice receiving various treatments including FCA only, free DIC, free DEX, or HA-Lipo-DIC/DEX after NE680 injection and their distribution in the paw was assessed by Near-infrared fluorescence-imaging. (**B**) Fluorescence activities of arthritic mice in various treated groups (mean ± SEM). Mice receiving free DEX or HA-Lipo-DIC/DEX exhibited the least fluorescence in the first 2 weeks. Four weeks after mice receiving FCA induced arthritic, the HA-Lipo-DIC/DEX treated group exhibited the least fluorescence imaging (^★^ indicates *p* < 0.05, ^★★^ indicates *p* < 0.001, one-way ANOVA).

**Figure 6 ijms-22-00665-f006:**
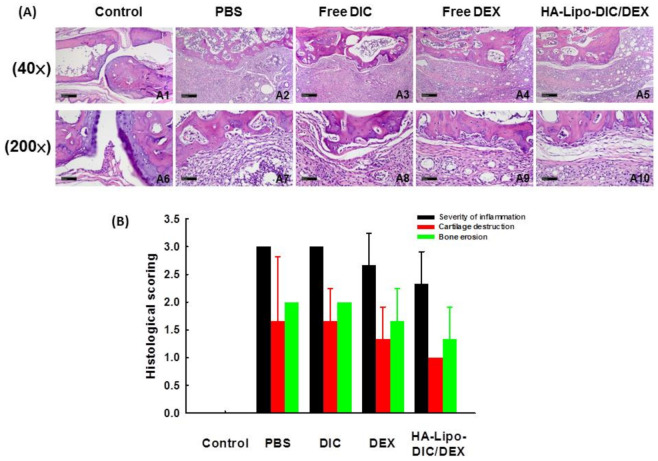
HA-Lipo-DIC/DEX inhibited leukocyte infiltration and cartilage destruction in OA-induced mice. (**A**) Hematoxylin and eosin staining revealed the histological changes in the various anti-inflammation formulation treatments including control (A1 and A6), phosphate-buffered saline (PBS) (A2 and A7), DIC (A3 and A8), DEX (A4 and A9), and HA-Lipo-DIC/DEX (A5 and A10) on day 28, respectively. (Scale bar (A1~A5) 40×: 200 μm; (A6~A10) 200×: 50 μm). (**B**) Arthritic changes, including inflammatory cell infiltration, synovial proliferation/enlargement, articular cartilage breakdown, and bone destruction, were evaluated 28 days after administration of Freund’s Complete Adjuvant (CFA), graded from 0 to 3, and summed. Results are presented as the mean ± SD of 3 animals/group.

**Figure 7 ijms-22-00665-f007:**
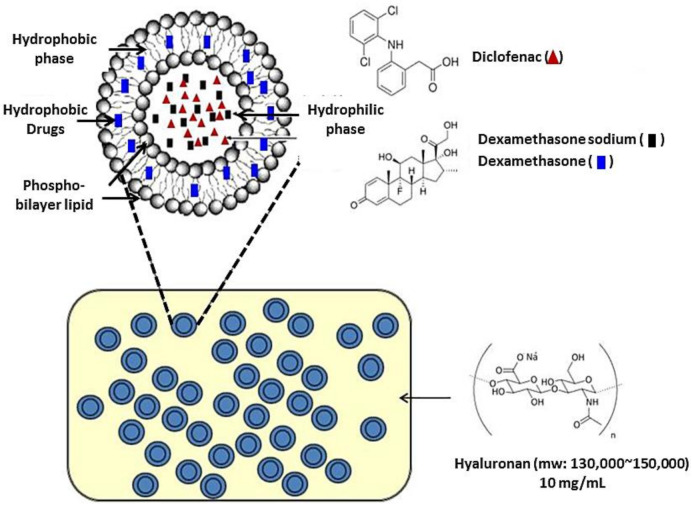
The schematic diagram of HA-Lipo-DIC/DEX structure. Liposomal nanoparticle Lipo-DIC/DEX containing both hydrophilic/hydrophobic DEX and DIC was first prepared. The preparation procedure was as described previously. HA-Lipo-DIC/DEX was prepared by mixing equal volume of liquefied HA (20 mg/mL) and Lipo-DIC/DEX as described in Materials and Methods Section. Liposomal nanoparticle structure was originally obtained from the website: https://www.slideshare.net/SunealSaini/liposomes-47868772 and modified by author. The final liquefied HA concentration was determined as 10 mg/mL.

**Table 1 ijms-22-00665-t001:** Characteristics of various dosages of DIC/DEX-loaded liposomes.

DIC (mg)	DEX (mg)	DEX Sodium (mg)	Encapsulation Efficiency (%) ^a^	DIC Content (%) ^b^	DEX Content (%) ^b^	Mean Size nm (PDI) ^c^	Zeta Potential (mV)
40	20	0	94.7 ± 2.4%	15.8 ± 2.1%	0.9 ± 0.3%	85.9 ± 0.3 (0.229)	−8.8 ± 3.9
40	15	5	93.1 ± 4.2%	16.4 ± 1.0%	1.6 ± 0.4%	98.5 ± 0.3 (0.236)	−25.5 ± 4.5
40	10	10	91.6 ± 2.7%	17.2 ± 2.4%	2.6 ± 1.0%	103.0 ± 0.2 (0.240)	−27.9 ± 5.4
40	5	15	92.4 ± 4.0%	16.1 ± 3.8%	4.4 ± 1.2%	92.4 ± 0.3 (0.191)	−16.4 ± 3.2
40	0	20	91.4 ± 3.5%	18.2 ± 3.2%	4.8 ± 1.5%	97.4 ± 0.2 (0.213)	−17.9 ± 4.4
40	10	0	92.1 ± 4.1%	15.1 ± 2.3%	1.2 ± 0.5%	100.0 ± 0.3 (0.205)	−27.1 ± 4.8
40	5	5	90.5 ± 5.6%	22.5 ± 4.1%	2.5 ± 0.6%	103.6 ± 0.3 (0.260)	−22.3 ± 4.6
40	0	10	93.9 ± 3.3%	20.8 ± 5.4%	3.7 ± 1.1%	98.6 ± 0.3 (0.264)	−25.5 ± 5.0

^a^ DIC/DEX encapsulation efficiency (%) = (weight of DIC/DEX in the liposomal nanoparticle/weight of the feeding DIC/DEX) × 100%. ^b^ Drug content (%) = (weight of DIC/DEX)/(weight of DIC/DEX + weight of polymer) × 100%. ^c^ As determined by DLS after filtration through a 0.22-μm filter. Abbreviations: DLS, dynamic light scattering; DIC, diclofenac; DEX, dexamethasone; PDI, polydispersity index.

## Data Availability

The data presented in this study is contained within the article.

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
