# Peer review of "Hyaluronan-Loaded Liposomal Dexamethasone–Diclofenac Nanoparticles for Local Osteoarthritis Treatment"

_ijms, 2021, doi:10.3390/ijms22020665_

Round 1
Reviewer 1 Report
In this contribution by Chang and co-workers, the authors investigated liposomal dexamethasone–diclofenac nanoparticles mixed with HA for local osteoarthritis treatment. The results are kind of interesting and potentially attractive to the readership of IJMS. However, lots of information are missing, and It could be publishable in due course but these points below must be addressed prior to publication.
1) In the introduction, it is not clear why the authors would like to add HA in the study. The authors should make this point clearer.
2) The stability of liposomes as a carrier for the loaded drugs is of important. Authors should add such discussion in the introduction. The approach to stabilize the liposomes is by electrostatic repulsion, steric repulsion, and strong hydration. In this study, it seems electrostatic repulsion plays the main role (zeta potential). Several recent papers[1-3] should be cited.
[1] N. Rezaei, F. Mehrnejad, Z. Vaezi, M. Sedghi, S. M. Asghari, H. Naderi-Manesh, Colloids and Surfaces B: Biointerfaces 2020, 185, 110552.
[2] W. Lin, N. Kampf, R. Goldberg, M. J. Driver, J. Klein, Langmuir 2019, 35, 6048.
[3] F. Mastrotto, C. Brazzale, F. Bellato, S. De Martin, G. Grange, M. Mahmoudzadeh, A. Magarkar, A. Bunker, S. Salmaso, P. Caliceti, Molecular pharmaceutics 2019, 17, 472.
3) There is no PDI information in table 1.
4) What’s the size and PDI after HA added to DIC/DEX-loaded liposomes?
5) In section 4.2, no information about the molecular weight of HA is provided.
6) In section 4.3, the DIC/DEX-loaded liposomes were prepared in pure water, but the drug release was done in PBS, there is huge difference in the osmotic pressure between inside and outside of liposomes, which will make the rapture of the liposomes (unstable). If the authors did this, the release profile, even the following animal experiments are not reasonbale. Please comment on it.
7) The authors should add a schematic to show where the HA molecule locates after being added to DIC/DEX-loaded liposomes. Outside the out leaflet of liposome?
Author Response
Reference No: ijms-1060567
TI: Hyaluronan-loaded liposomal dexamethasone–diclofenac nanoparticles for local osteoarthritis treatment
Reviewer #1 (Comments to the Author):
In this contribution by Chang and co-workers, the authors investigated liposomal dexamethasone–diclofenac nanoparticles mixed with HA for local osteoarthritis treatment. The results are kind of interesting and potentially attractive to the readership of IJMS. However, lots of information are missing, and It could be publishable in due course but these points below must be addressed prior to publication.
Answer: Thank you for your comments.
1) In the introduction, it is not clear why the authors would like to add HA in the study. The authors should make this point clearer.
Answer: Thank you for your comments. We have added a paragraph about HA to make our point clear. (Please see Page 7, Para 3)
2) The stability of liposomes as a carrier for the loaded drugs is of important. Authors should add such discussion in the introduction. The approach to stabilize the liposomes is by electrostatic repulsion, steric repulsion, and strong hydration. In this study, it seems electrostatic repulsion plays the main role (zeta potential). Several recent papers[1-3] should be cited.
[1] N. Rezaei, F. Mehrnejad, Z. Vaezi, M. Sedghi, S. M. Asghari, H. Naderi-Manesh, Colloids and Surfaces B: Biointerfaces 2020, 185, 110552
[2] W. Lin, N. Kampf, R. Goldberg, M. J. Driver, J. Klein, Langmuir 2019, 35, 6048
[3] F. Mastrotto, C. Brazzale, F. Bellato, S. De Martin, G. Grange, M. Mahmoudzadeh, A. Magarkar, A. Bunker, S. Salmaso, P. Caliceti, Molecular pharmaceutics 2019, 17, 472.
Answer: Thank you for your comments. We have added a paragraph to describe the importance of zeta potential in stabilizing the liposomes. (Please see Page 6, Para 2, Line 4 to Line 11). And we also add these three references to our manuscript. (Please see Reference 18-20)
3) There is no PDI information in table 1
Answer: Thank you for your comments. We have added the PDI data in Table 1 (Please see Table 1).
4) What’s the size and PDI after HA added to DIC/DEX-loaded liposomes?
Answer: Thank you for your comments. For HA is mixed with Lipo-DIC/DEX to form the gel-like osteoarthritis therapeutic formulation, we apologize that it is difficult to analysis the size and PDI after HA added to DIC/DEX-loaded liposomes in our lab at present. However, we will cooperate with the other research teams to test if the size and PDI of HA-Lipo-DIC/DEX could be analyzed.
5) In section 4.2, no information about the molecular weight of HA is provided
Answer: Thank you for your comments. The molecular weight of HA we used to prepare HA-Lipo-DIC/DEC is around 130,000-150,000. And we have added the information to our manuscript (Please see Page 9, Para 2, Line 1)
6) In section 4.3, the DIC/DEX-loaded liposomes were prepared in pure water, but the drug release was done in PBS, there is huge difference in the osmotic pressure between inside and outside of liposomes, which will make the rapture of the liposomes (unstable). If the authors did this, the release profile, even the following animal experiments are not reasonbale. Please comment on it
Answer: Thank you for your comments. We apologized for we don’t make our liposome preparation procedure clear. As described in “Materials and Methods” section, the liposomal system was filtered by 0.2 μM filter. The filtered liposome will be further purified by PD-10 column by using 50°C PBS to elute the liposome. We have rewritten the sentence. (Please see Page 10, Para 1, Line 2 to Line 5)
7) The authors should add a schematic to show where the HA molecule locates after being added to DIC/DEX-loaded liposomes. Outside the out leaflet of liposome?
Answer: Thank you for your comments. We have added the figure 7 to illustrate the structure of HA-Lipo-DIC/DEX. As we demonstrated, the HA was mixed with liposomal nanoparticle Lipo-DIC/DEX. (Please see Figure 7)
Reviewer 2 Report
It is appropriate to report the molecular weight of the HA utilized.
Report reference for the statement on line 174 and update reference 32.
Author Response
Reviewer #2 (Comments to the Author):
1) It is appropriate to report the molecular weight of the HA utilized.
Answer: Thank you for your comments. The molecular weight of HA we used to prepare HA-Lipo-DIC/DEC is around 130,000-150,000. And we have added the information to our manuscript (Please see Page 9, Para 2, Line 1)
2) Report reference for the statement on line 174 and update reference 32.
Answer: Thank you for your comments. We have added two references for HA topically application in OA treatment in clinical trials. (Please see Page 20, Para 1, Line 2 and Reference 35-36). We also deleted the original reference 32 (An evidence-based update on nonsteroidal anti-inflammatory drugs. Clin. Med. Res. 2007, 5, 19-34) and added a new reference (Topical Diclofenac, an Efficacious Treatment for Osteoarthritis: A Narrative Review. Rheumatol. Ther. 2020, 7, 217-236). (Please see Reference 37)

Round 2
Reviewer 1 Report
The authors did a serious revision, and solved most of the issues. After moving the ref. [18,19] to line 78 instead of line 80, I recommend it for publication in IJMS.